# Comparative Study of α- and β-MnO_2_ on Methyl Mercaptan Decomposition: The Role of Oxygen Vacancies

**DOI:** 10.3390/nano13040775

**Published:** 2023-02-19

**Authors:** Hong Su, Jiangping Liu, Yanan Hu, Tianhao Ai, Chenhao Gong, Jichang Lu, Yongming Luo

**Affiliations:** 1Faculty of Environmental Science and Engineering, Kunming University of Science and Technology, Kunming 650500, China; 2The Innovation Team for Volatile Organic Compounds Pollutants Control and Resource Utilization of Yunnan Province, The Higher Educational Key Laboratory for Odorous Volatile Organic Compounds Pollutants Control of Yunnan Province, Kunming 650500, China; 3Faculty of Chemical Engineering, Kunming University of Science and Technology, Kunming 650500, China

**Keywords:** CH_3_SH decomposition, MnO_2_, oxygen vacancies, oxygen migration

## Abstract

As a representative sulfur-containing volatile organic compounds (S-VOCs), CH_3_SH has attracted widespread attention due to its adverse environmental and health risks. The performance of Mn-based catalysts and the effect of their crystal structure on the CH_3_SH catalytic reaction have yet to be systematically investigated. In this paper, two different crystalline phases of tunneled MnO_2_ (α-MnO_2_ and β-MnO_2_) with the similar nanorod morphology were used to remove CH_3_SH, and their physicochemical properties were comprehensively studied using high-resolution transmission electron microscope (HRTEM) and electron paramagnetic resonance (EPR), H_2_-TPR, O_2_-TPD, Raman, and X-ray photoelectron spectroscopy (XPS) analysis. For the first time, we report that the specific reaction rate for α-MnO_2_ (0.029 mol g^−1^ h^−1^) was approximately 4.1 times higher than that of β-MnO_2_ (0.007 mol g^−1^ h^−1^). The as-synthesized α-MnO_2_ exhibited higher CH_3_SH catalytic activity towards CH_3_SH than that of β-MnO_2_, which can be ascribed to the additional oxygen vacancies, stronger surface oxygen migration ability, and better redox properties from α-MnO_2_. The oxygen vacancies on the catalyst surface provided the main active sites for the chemisorption of CH_3_SH, and the subsequent electron transfer led to the decomposition of CH_3_SH. The lattice oxygen on catalysts could be released during the reaction and thus participated in the further oxidation of sulfur-containing species. CH_3_SSCH_3_, S^0^, SO_3_^2−^, and SO_4_^2−^ were identified as the main products of CH_3_SH conversion. This work offers a new understanding of the interface interaction mechanism between Mn-based catalysts and S-VOCs.

## 1. Introduction

As a particular class of volatile organic compounds (VOCs), sulfur-containing volatile organic compounds (S-VOCs) can be converted into sulfate aerosols in the atmosphere through complex physicochemical reactions [1]. They can also react indirectly with NO_x_ through photochemistry reactions, which are the crucial precursors for forming PM _2.5_ and O_3_. Methyl mercaptan (CH_3_SH), a representative S-VOC, is considered as an important air odor pollutant, which is harmful to the ecosystem and human health owing to its severe toxicity and low olfactory threshold [2,3,4]. In previous studies, various methods have been employed to eliminate CH_3_SH, such as adsorption [5,6], biodegradation [7], photocatalytic oxidation [8], and catalytic oxidation [9,10]. However, these remediation technologies suffer from secondary pollution because of incomplete removal and high cost. Until now, catalytic decomposition has been regarded as the most promising strategy for removing S-VOCs due to its high purification efficiency, energy-saving nature, lack of additional additives (O_2_, H_2_, O_3,_ etc.), and less secondary pollution [11,12].

Metal-based catalysts are widely used for the removal of VOCs on account of their superior catalytic performance. Among the various transition metal oxides, MnO_2_ is of great interest because of its low cost, low toxicity, environmental friendliness, and wide natural distribution [13,14]. Meanwhile, MnO_2_ has been extensively studied in heterogeneous catalysis due to its unique physicochemical properties (i.e., multivalent, reactive oxygen species, and polycrystalline nature) and is recognized as one of the most active catalysts for VOCs removal among transition metal oxides [15,16,17]. Nanostructured MnO_2_ possesses a rich structural flexibility, which adopts various crystallographic forms such as α-, β-, δ-, and γ-MnO_2_, depending upon the size of the tunnel [18,19]. These polymorphs of MnO_2_ include a one-dimensional chain-like tunnel (α-, β-, and γ-MnO_2_) and two-dimensional layer structures (δ-MnO_2_) based on different linkage ways of the basic octahedral molecular sieves [20,21].

Among the MnO_2_ polymorphs, α-MnO_2_ has one of the largest tunnel sizes (4.6 Å) consisting of 1D (1 × 1) and (2 × 2) channels, composed of double chains of edge-sharing [MnO_6_] octahedra, filled with alkali metal cations, NH_4_^+^ or H_3_O^+^, inside the 2 × 2 tunnels to stabilize the structure [16]. The pyrolusite-type β-MnO_2_ has a rutile-type structure with infinite [MnO_6_] octahedral chains that share opposing edges. Each chain is connected to four similar chain corners, forming the smallest tunnel structure (2.3 Å) of MnO_2_ polymorphs, consisting of 1D (1 × 1) and (1 × 1) channels [22,23]. It is generally accepted that catalysts with different crystal structures exhibit different catalytic efficiencies and reaction mechanisms for the reactants. For instance, Hayashi et al. evaluated the aerobic catalytic performance of six types of MnO_2_ (α-, β-, γ-, δ, λ, ε -phases) for the transformation of 5-hydroxymethylfurfural to 2,5-furandicarboxylic acid and concluded the best activity of β-MnO_2_ [19]. Chen et al. found that the α- and γ-MnO_2_ presented higher benzene oxidation activity than β- and δ-MnO_2_, whereas δ-MnO_2_ displayed the best in formaldehyde oxidation among all of the MnO_2_ materials [20]. This may be due to the varying oxygen species in different types of MnO_2_, which play distinct roles in the catalytic oxidation of formaldehyde and benzene. In addition, surface defects of manganese-based catalysts are regarded as an important determinant of their catalytic activity, and their formation and driving of catalytic reactions are often related to their surface oxygen species [24,25,26]. Yang et al. investigated the phase-activity relationship of MnO_2_ toward toluene catalytic oxidation. They proposed that the excellent catalytic performance of δ-MnO_2_ may be associated with the rich oxygen vacancy and the strong mobility of oxygen species [27]. Tian et al. prepared α-, β-, and ε- MnO_2_ for CO oxidation and found that β-MnO_2_ possesses the lowest energies for oxygen vacancy generation as well as excellent redox properties, thus exhibiting the best CO oxidation activity [13]. These studies highlight the importance of oxygen vacancies in VOC removal and illustrate that the concentration of oxygen vacancies in different crystal structures tends to dominate their catalytic activity. Therefore, it is necessary to understand the relationship between the catalytic activity of manganese oxide and its surface structure to provide a standard for the further modification of MnOx or other metal oxide catalysts. However, noticeable differences are presented in the surface morphology and crystal structure of different crystalline MnO_2_, which makes it difficult to clarify the contribution of oxygen vacancies to the catalytic reaction. For example, δ-MnO_2_ is a layered structure, whereas α-, β- and γ-MnO_2_ are common tunneling structures. Among them, γ-MnO_2_ is a spherical structure, and α- and β-MnO_2_ are similar nanorod-like structures. The differences in crystal structure and surface morphology can inherently lead to variations in catalyst surface properties, which can obscure the critical role of oxygen vacancies. Therefore, it is essential to reveal the effect of surface vacancies on catalytic reactions based on the same morphology. In addition, oxygen vacancy-mediated catalytic reactions may be accompanied by the migration and release of oxygen species and the generation of new oxygen vacancies, thus the transformation of these active surface species in catalytic reactions and their contribution to the removal of VOCs need further clarification.

Herein, we compared the removal efficiency of CH_3_SH by α-MnO_2_ and β-MnO_2_ with similar surface morphology but different crystal structures. Their physicochemical properties were subsequently characterized by various analysis techniques. The number of surface low valence Mn, oxygen vacancies and redox properties were studied regarding high-resolution transmission electron microscope (HRTEM) and electron paramagnetic resonance (EPR), X-ray photoelectron spectroscopy (XPS), H_2_-TPR and O_2_-TPD. The changes of catalyst surface species before and after the reaction were characterized by XPS, and the variation of intermediate species of CH_3_SH during the reaction were also monitored.

## 2. Materials and Methods

### 2.1. Chemical Reagents

α-MnO_2_ and β-MnO_2_ were synthesized through the hydrothermal synthesis method according to previous research [28]. Potassium permanganate (KMnO_4_, Chengdu Colon Chemicals Co., LTD, Chengdu, China), hydrated manganese sulfate (MnSO_4_·H_2_O, Aladdin Reagent Co., LTD, Shanghai, China) and ammonium persulfate ((NH_4_)_2_S_2_O_8_, Aladdin Reagent Co., LTD, Shanghai, China) were used without further purification.

Synthesis of α-MnO_2_: 0.1 M KMnO_4_ and 0.05 M MnSO_4_·H_2_O were dissolved in 70 mL deionized water and stirred for 30 min. The resulting solution was transferred to a 100 mL Teflon-lined autoclave and maintained at 160 °C for 12 h. After cooling to room temperature, the precipitate was centrifuged and washed with distilled water (700–1000 mL) three times. Finally, the precipitate was dried at 80 °C for 4 h and calcination at 360 °C for 2 h.

Synthesis of β-MnO_2_: 0.14 M MnSO_4_·H_2_O and 0.14 M (NH_4_)_2_S_2_O_8_ were dissolved in 70 mL deionized water and stirred for 30 min. The resulting solution was transferred to a 100 mL Teflon-lined autoclave and maintained at 140 °C for 12 h. After cooling to room temperature, the precipitate was centrifuged and washed with distilled water (700–1000 mL) three times. Finally, the precipitate was dried at 80 °C for 4 h and calcination at 360 °C for 2 h.

### 2.2. Catalyst Characterization

The refined test of X-ray powder diffraction (XRD) of the products was performed using a Bragg-Brentano-type powder diffractometer (Nihongo TTRIII, Tokyo City, Japan, operated at 40 kV and 200 mA, Cu Kα radiation, λ = 0.15418 nm). To investigate the Brunauer-Emmett-Teller (BET) surface areas, average pore diameters, and total pore volumes of the samples, N_2_ adsorption-desorption isotherms were determined using a NOVA 4200e Surface Area and Pore Size Analyzer. Electron paramagnetic resonance (EPR) signals were carried out on a Bruker A300 spectrometer (Saarbrucken, Germany) at 25 °C. XPS profiles were obtained with a Thermo Scientific K-Alpha spectrometer (Waltham, MA, USA). The binding energy (BE) values were calibrated using the C 1 s peak at 284.8 eV. The Raman spectra were recorded using a 514 nm laser excitation source with an integration time of 3 s and 30 accumulations (Raman, BX41, HOEIBA Scientific, Paris, France). Scanning electron microscopy (SEM, VEGA3SBH, Brno, Czech Republic) and high-resolution transmission electron microscopy (HRTEM, Talos F200X, Thermo Scientific, Waltham, MA, USA) were used to observe catalyst morphologies.

Hydrogen temperature-programmed reduction (H_2_-TPR) and oxygen temperature- programmed desorption (O_2_-TPD) experiments were performed on a FULI II 7970 gas chromatograph (Fuli Analytical Instrument Inc., Hangzhou, China) with a thermal conductivity detector (TCD). In H_2_-TPR experiments, 50 mg of the sample was placed in a quartz tube and pretreated in a gas flow of 10% H_2_/Ar (30 mL min^−1^) at 100 °C for 30 min to remove impurities. After the pretreatment process, the sample was reduced by 10% H_2_/Ar (30 mL min^−1^) from 100 to 800 °C with a heating rate of 10 °C/min. For O_2_-TPD analysis, 50 mg of sample was loaded on the quartz tube, heated to 105 °C and pretreated with He (30 mL min^−1^) for 30 min to remove surface adsorbed water, followed by cooling to 30 °C. Subsequently, the sample was adsorbed by 10% O_2_/He (30 mL min^−1^) at room temperature for 60 min, and then He (30 mL min^−1^) was used to purge the sample for 30 min to remove physically adsorbed O_2_ and stabilize the baseline. Subsequently, the temperature was ramped from 30 to 850 °C at 10 °C mL min^−1^.

### 2.3. Catalyst Activity Evaluation

The catalytic performance for the CH_3_SH decomposition was investigated in a fixed-bed quartz tube reactor (i.d. = 6 mm). 200 mg samples with the size of 40–60 meshes were loaded into the reactor. The reaction temperature was controlled and maintained for about 1 h at each designated temperature. The inlet CH_3_SH concentration was set at 5000 ppm, and the total flow rate was maintained at 30 mL min^−1^. The concentration of CH_3_SH was recorded by GC-9790 (FULI, China) equipped with a flame ionization detector (FID) and flame photometric detector (FPD), and the CH_3_SH conversion ratio was calculated as follows:$${\mathrm{CH}}_{3}\mathrm{SH}\;\mathrm{conversion}=\frac{C_{in}-C_{out}}{C_{in}}\times 100\%$$

$C_{in}\;$ represents the inlet concentration of CH_3_SH and $C_{out}$ is the outlet concentration of CH_3_SH.

The reaction rates of CH_3_SH decomposition were determined in the kinetic regime at a CH_3_SH conversion lower than 20% at different temperatures; The reaction rate (r_CH3SH_; mol g^−1^ h^−1^) for CH_3_SH decomposition was calculated according to the following equations:$$r_{CH_{3}SH}=\frac{C_{CH_{3}SH}\times X_{CH_{3}SH}\times F}{m_{cat}}$$
$$r_{norm}=\frac{C_{CH_{3}SH}\times X_{CH_{3}SH}\times F}{m_{cat}\times S_{BET}}$$
where the $\;C_{CH_{3}SH}\;$ represents the initial methyl mercaptan concentration, F (mol·h^−1^) represents the total flow rate, $X_{CH_{3}SH}\;$ denotes CH_3_SH conversion, and $S_{BET}$ (m^2^·g^−1^) represents the specific surface area of catalysts.

The turnover frequency (TOF, h^−1^) was calculated for different crystal types based on oxygen vacancy concentration for MnO_2_, and indicates the number of reactions of methyl mercaptan at each active site per unit of time, thus TOF was obtained using the following equation:$$TOF\left(h^{-1}\right)=\frac{C_{CH_{3}SH}\times X_{CH_{3}SH}\times F}{\frac{m_{MnO_{2}}}{M_{MnO_{2}}}\times \left(Mn^{2+}+Mn^{3+}\right)}$$

$M_{MnO_{2}}$ (mol·g^−1^) is the molar mass of $MnO_{2}$, and $Mn^{2+}+Mn^{3+}$ derived from XPS data, which represent the concentration of the oxygen vacancies of MnO_2_ deduced from the obtained XPS spectra.

## 3. Results and Discussion

### 3.1. Structure and Morphology

XRD was used to determine the crystal structure of the prepared material. The XRD patterns of as-prepared MnO_2_ with various crystal types are shown in Figure 1. The diffraction peaks located at ~12.7°, ~18.1°, ~28.8°, and ~37.5° can be assigned to α-MnO_2_ (JCPDS card no. 44-0141) (Figure 1A), and the peaks at ~28.7°, ~37.2°, ~42.7° and ~56.4° can be ascribed to β-MnO_2_ (JCPDS card no. 24-0735) (Figure 1B) [29]. There was no apparent crystal transformation on these samples after 360 °C calcination. The sharp and strait peaks of β-MnO_2_ could indicate its great crystallization and large grain size; α-MnO_2_ presented wide bands with relatively lower crystallinity and smaller grain sizes. The above results indicate the successful obtaining of the two kinds of MnO_2_ with specific crystal phases.

The morphologies of MnO_2_ samples were characterized by a scanning electron microscopy (SEM), a transmission electron microscope (TEM), and a high-resolution TEM (HRTEM). As can be seen in Figure 2A, α-MnO_2_ showed a stacking-nanorod structure with an average length of about 330 nm. β-MnO_2_ (Figure 2F) also showed a typical rod shape with a diameter near 50 nm and a length of ~1.2 μm. TEM showed consistent results with the SEM that α-MnO_2_ (Figure 2B) and β-MnO_2_ (Figure 2G) had similar nanorod-like structures as previously reported [30]. The well-identified periodic lattice fringes of 6.9 Å can be clearly observed in Figure 2C, corresponding to the interplanar distance of (110) facet of α-MnO_2_. Figure 2H exhibited the lattice fringes of 3.1 Å, which match the interplanar distance of the (110) facet of β-MnO_2_ well. Compared with the β-MnO_2_ samples, α-MnO_2_ showed more blurry lattice fringes, representing poor crystallinity, which also agrees well with the XRD patterns. In addition, the presence of defects was further demonstrated using the inverse Fast Fourier Transform (FFT) pattern (Figure 2E,J). Significantly more lattice distortion can be clearly observed on the surface of α-MnO_2_ (Figure 2D) (highlighted by red ovals), thus leading to more defects than β-MnO_2_ [10]. Besides, severe blurring of the lattice fringes was also detected on α-MnO_2_ than β-MnO_2_. Lattice distortion can be caused by nearby point defects. Simultaneously, a defect layer will be formed once the defect concentration is high enough, resulting in a blurry lattice fringe in the HRTEM images [31]. Hence, the intrinsic defective structure of α-MnO_2_ was confirmed. Oxygen vacancies, as an important point defect in catalysts, play a prominent role in the catalytic reaction process, and the high oxygen vacancy concentration will result in a blurry lattice fringe, which can be reflected in the HRTEM images [32]. As shown in Figure 3, the EPR signal corresponding to g = 2.003 can be attributed to oxygen vacancies, and its signal intensity can represent the number of oxygen vacancies [27]. Therefore, more oxygen vacancies on α-MnO_2_ than β-MnO_2_ can be confirmed based on EPR, consistent with HRTEM analysis.

The BET surface areas (S_BET_), and pore volumes of the two catalysts are shown in Figure 4A,B. It is reported that the different structures assembled by MnO_6_ octahedra in MnO_2_ will affect the related surface areas and pore volumes. β-MnO_2_ presented relatively low specific surface areas (12.76 m^2^ g^−1^) and pore volumes (0.06 cm^3^ g^−1^), whereas α-MnO_2_ showed higher specific surface areas (34.59 m^2^ g^−1^) and pore volumes (0.13 cm^3^ g^−1^). Moreover, the nitrogen adsorption-desorption isotherms of α-MnO_2_ and β-MnO_2_ displayed a type IV curve with H_3_-type hysteresis loops, indicating that both samples were mesoporous structures [33].

### 3.2. Catalytic Performance

In order to explore the activity of two catalysts on sulfur-containing volatile organic pollutants (S-VOCs), methyl mercaptan (CH_3_SH) was chosen as the model S-VOCs, and the catalytic activities of α-MnO_2_ and β-MnO_2_ are shown in Figure 5A. Two MnO_2_ samples exhibited significantly different catalytic performance in CH_3_SH catalytic reaction. α-MnO_2_ (74%) exhibited significantly better catalyst activity than β-MnO_2_ (3%) at 30 °C. The decreases of CH_3_SH conversion for α-MnO_2_ at 50 °C may be due to the desorption of CH_3_SH on the catalyst. As the temperature increased, the conversion of CH_3_SH reached 100% at 100 °C with both catalysts. Furthermore, the reaction rates of α-MnO_2_ and β-MnO_2_ at 50 °C were calculated based on the activity experiments. As shown in Figure 5B, α-MnO_2_ showed the CH_3_SH reaction rate of 2.9 × 10^−2^ mol g^−1^ h^−1^, this being ~4.1 times higher than the rates measured for and β-MnO_2_ at 50 °C, which was consistent with the results for the catalytic activity.

It is well known that the specific surface area plays a critical role in catalytic reactions. To eliminate its influence, the reaction rates with surface area normalization were calculated at different temperatures based on the data from the activity experiments. The results of the normalized reaction rates (r_norm_, mol m^−2^ h^−1^) of CH_3_SH decomposition are shown in Figure 6A. The normalized reaction rates for α-MnO_2_ were obviously higher than those of β-MnO_2_ at 30, 50, 60 and 80 °C, which suggested that reactivity was not governed by the specific surface area. Turnover frequency (TOF) is essential for studying the intrinsic reactivity of catalysts. In this work, the TOF (h^−1^) was calculated based on oxygen vacancy concentration, and the TOF value for the CH_3_SH catalytic decomposition was conducted at 50 °C with 0.01 g of catalyst and was calculated within a low CH_3_SH conversion (1 h of reaction, below 15.0%). As displayed in Figure 6B, the α-MnO_2_ showed the highest TOF value of 0.14 h^−1^, which was 1.8 times as that of the β-MnO_2_ (0.08 h^−1^), indicating that α-MnO_2_ has better catalytic performance for CH_3_SH.

### 3.3. Redox Capacity and Oxygen Species

To evaluate the reduction behaviors of MnO_2_ samples, H_2_-temperature-programmed reduction (TPR) was performed (Figure 7A). For α-MnO_2_, the peaks at 289 and 309 °C corresponded to the reduction of Mn^4+^ → Mn^3+^ and Mn^3+^ → Mn^2+^, respectively, and the peaks around 291 and 317 °C for β-MnO_2_ were attributed to Mn^4+^ → Mn^3+^ and Mn^3+^ → Mn^2+^, respectively [20,34]. The reduction temperature of α-MnO_2_ was lower than that of β-MnO_2_, indicating that the reduction of α-MnO_2_ is relatively faster. More importantly, the more remarkable reduction ability of α-MnO_2_ means easier deoxygenation during hydrogen treatment, suggesting that oxygen migration is more likely to occur on its surface. Therefore, α-MnO_2_ features stronger oxygen species mobility than β-MnO_2_.

O_2_-TPD was conducted further to explore the oxygen species of the MnO_2_ catalysts. Figure 7B shows three desorption peaks related to oxygen species that can be observed on the MnO_2_. The low-temperature peak below 400 °C was ascribed to the chemisorbed active oxygen species on the surface (O^−^ and O_2_^−^) [35]. The desorption peaks at 400–650 °C and 700–850 °C were related to the release of subsurface and bulk lattice oxygen species (O^2−^), respectively [36,37]. The desorption of surface oxygen at low temperature (<400 °C) plays the primary role as reactive oxygen species participating in the catalytic reaction [38]. Moreover, the lower temperature of the surface oxygen desorption peak means better low-temperature mobility of oxygen species. As depicted in Figure 7B, α-MnO_2_ showed a lower temperature at 146 °C of the surface oxygen desorption peak than β-MnO_2_ at 367 °C, indicating the better low-temperature mobility of oxygen species, which is in agreement with H_2_-TPR.

More bonding properties were discussed through Raman spectra (Figure 8A). The peaks at 348 and 640 cm^−1^ corresponded to the Mn-O bending and the stretching vibration, respectively [39]. Significantly weaker and broader Raman peaks at around 640 cm^−1^ were detected for α-MnO_2_ than β-MnO_2_, suggesting lower crystallinity and more defects due to the lattice distortion [40]. To evaluate the strength of the Mn-O bond, the bond force constant (k) was calculated from Hooke’s law [41,42] using the following equation: **ω =**$\;\frac{1}{2\Pi c}\sqrt{\frac{k}{\mu }}$**,** where ω is the Raman shift (cm^−1^), c is light velocity, and μ is the effective mass of the Mn-O bond. The calculated Mn-O force constant (k) is shown in the inset of Figure 8B. Thus, the Mn-O bond force constant of α-MnO_2_ (293 N/m) was smaller than that of β-MnO_2_ (296 N/m), implying the weaker Mn-O bond. The weaker Mn-O bond means easier migration of O and easier redox of Mn during the reaction, which is beneficial for catalytic reactions [25].

### 3.4. Identification of the Role of Oxygen Vacancies in CH_3_SH Degradation

The surface elemental composition and chemical state of these MnO_2_ samples were identified by XPS. The XPS spectra of Mn 2p_3/2_ of the samples are shown in Figure 9. The peaks corresponding to binding energies at 642.7, 641.7 and 640.6 eV can be attributed to Mn^4+^, Mn^3+^ and Mn^2+^, respectively [43,44]. It is noteworthy that the binding energy corresponding to different valences of Mn were slightly different in both MnO_2_ samples, indicating that crystal phase structure has a certain effect on the electron density of the MnO_2_ surface, which is related to the degree of charge imbalance, oxygen vacancies, as well as the relative content of Mn^2+^, Mn^3+^ and Mn^4+^ [45]. Specifically, the oxygen vacancy will be generated to maintain electrostatic balance with the increasing Mn^2+^ and Mn^3+^ proportion, and the proportion of low-valence Mn is generally regarded as an indicator of surface oxygen vacancies [44]. As shown in Table 1 and Figure 9A, the proportion of the low valence Mn (Mn^2+^ + Mn^3+^) showed α-MnO_2_ (41.62%) > β-MnO_2_ (37.74%). Besides, the average oxidation state (AOS) of MnO_2_ was calculated according to the formula of AOS = 8.956 − 1.126 ΔE [46], which was based on the size of Mn 3s multiple splitting (ΔE) in Mn 3s XPS spectra (Figure 10A). The AOS values of the Mn element in α-MnO_2_ and β-MnO_2_ were calculated to be 3.57 and 3.75, respectively. In previous studies, lower AOS of MnO_2_ was also able to indicate more surface oxygen vacancies [47]. Therefore, it can be inferred that α-MnO_2_ has a greater surface oxygen vacancy density than β-MnO_2_ (in agreement with HETEM and EPR).

Additionally, Mn^2+^-O and Mn^3+^-O bonds are weaker than Mn^4+^-O [48]. Thus, the higher ratio of low valence Mn (Mn^2+^ + Mn^3+^) endows MnO_2_ with a larger proportion of weaker M-O bonds on its surface, meaning the easier release of O to participate in the reaction. Figure 9A showed that α-MnO_2_ had more low valence Mn (Mn^2+^ + Mn^3+^) than β-MnO_2_, which also implies the easier release of surface oxygen species, consistent with the results of H_2_-TPR and O_2_-TPD. Comparison of the Mn 2p_3/2_ spectra before and after the reaction (Figure 9B) of MnO_2_ with CH_3_SH showed that the valence state of Mn in both samples changed obviously, suggesting the electron transfer during the reaction. For both samples, Mn^4+^ decreased, and Mn^2+^ and Mn^3+^ increased, proving that the high-valent Mn (IV) was reduced by gaining electrons during the reaction. After the reaction, the Mn^2+^+Mn^3+^/Mn^4+^ of α-MnO_2_ increased by 1.51 and that of β-MnO_2_ by 1.39, and AOS decreased by 0.74 for α-MnO_2_ and that of β-MnO_2_ decreased by 0.5 (Figure 10B), testifying that α-MnO_2_ was reduced to a greater extent by gaining more electrons than β-MnO_2_, which can well match H_2_-TPR results. It is noteworthy that higher AOS usually indicates a stronger electron-gaining ability of the catalyst because of the presence of more high-valent atoms, however, α-MnO_2_ exhibited a stronger electron-gaining ability in the reaction with CH_3_SH, suggesting that oxygen vacancies play a more important role in catalyzing CH_3_SH comparing to the high-valent Mn. This may explain the fact that chemisorption is the rate-limiting step for electron transfer, and more surface oxygen vacancies provide more surface adsorption sites for CH_3_SH.

In addition, the reaction between CH_3_SH and the catalyst could change the electronic environment of the catalyst. During the reaction, as the ratio of Mn^2+^ and Mn^3+^ increased, weaker Mn-O bonds were continuously formed and broken, leading to deoxygenation and further generation of oxygen vacancies to maintain electrostatic equilibrium, which may provide new sites for the reaction, and these desorbed oxygen species may favor the catalytic oxidation of CH_3_SH as well.

The XPS spectra of O 1s of the samples are shown in Figure 11. Peaks with binding energies at 529–529.8, 530.9–532 and 533 eV in the XPS spectra of O 1s of the MnO_2_ samples (Figure 11) can be attributed to lattice oxygen (O_latt_) and surface adsorption oxygen (O_ads_), and surface hydroxyl oxygen (O_adsO-H_), respectively [49,50,51]. The molar ratio of O_ads_/O_latt_ is shown in Table 2 and follows the order of α-MnO_2_ (0.65) > β-MnO_2_(0.35). O_ads_ was generally considered the most reactive oxygen species in the catalytic reaction and capable of participating in the catalytic oxidation of VOCs in previous reports [52,53]. The oxygen species changes before and after the reaction are shown in Figure 11B and Table 2. The O_latt_ and O_ads_ for both two materials decreased and increased, respectively, suggesting the migration of O_latt_ to form O_ads_ during the reduction of Mn. Obviously, α-MnO_2_ formed more O_ads_ after the reaction, which corresponds to its greater degree of reduction, also implying that α-MnO_2_ has a stronger catalytic capacity.

### 3.5. Product Detection during the Reaction

The main gas phase products were monitored quantitatively to better understand the reaction process of CH_3_SH over two different catalysts. As displayed in Figure 12, the decomposition of CH_3_SH at different temperatures corresponded to the production of CH_3_SSCH_3_. Meanwhile, the concentration of CH_3_SH during the reaction showed an excellent correlation with the concentration of CH_3_SSCH_3_, indicating that CH_3_SSCH_3_ was the main gas-phase product. At 150 °C, CH_3_SH was completely decomposed for both MnO_2_ catalysts, consistent with the thermodynamic theory that catalytic reactions proceed easier at higher temperatures. The yield of CH_3_SSCH_3_ gradually decreased when T > 100 °C, which may be due to the further catalytic oxidation of CH_3_SSCH_3_ at higher temperatures.

Figure 13 shows the changes of S 2p before and after the reaction, which was used to detect the solid-phase intermediates in the reaction process. No S species were detected on two catalysts before the reaction. In contrast, significant amounts of S species were detected on both samples after the reaction, indicating that some sulfur-containing products were adsorbed on the catalyst surface. As shown in Figure 13B, three peaks at 163.2, 167.9, and 169.2 eV corresponding to S^0^, SO_3_^2−^ (S^4+^), and SO_4_^2−^ (S^6+^), respectively [30,54], which all showed higher valence than S^2-^ from CH_3_SH, indicating the oxidation of S during the reaction. It is worth noting that the catalytic experiments in CH_3_SH were performed under a nitrogen atmosphere, so it can be concluded that the O in the S-O species was mainly derived from the O_ads_ of MnO_2_. This illustrated the O_ads_ involvement in the catalytic reaction of CH_3_SH. SO_3_^2−^ and SO_4_^2−^ were mainly retained on the manganese dioxide surface in the form of MnSO_3_ and MnSO_4_, implying that chemisorption was a prerequisite for the decomposition of CH_3_SH on MnO_2_. Notably, reacted α-MnO_2_ showed a higher proportion of SO_4_^2−^ (S^6+^) (26.31%) than β-MnO_2_ (12.56%) (Table 3 and Figure 13B), suggesting a greater degree of S oxidation, which corresponds to a greater reduction of Mn^4+^ after the reaction (Table 1 and Figure 9B). Furthermore, SO_4_^2−^ requires more oxygen to be coordinated with S than SO_3_^2−^, so a higher proportion of SO_4_^2−^ production requires more O_ads_ to participate in the reaction. Correspondingly, the H_2_-TPR, O_2_-TPD, and XPS analysis demonstrated more O_ads_ and better surface oxygen mobility for α-MnO_2_ than β-MnO_2_.

Based on the above experimental and characterization analysis, the catalytic mechanism of CH_3_SH by α-MnO_2_ and β-MnO_2_ can be inferred in Figure 14. CH_3_SH was first chemisorbed on the MnO_2_ surface and subsequently underwent a single electron transfer to form CH_3_S·, and then the two CH_3_S· were coupled to form CH_3_SSCH_3_. Based on the formation of S-S bonds, it is speculated that the single electron transfer occurs on S, suggesting that the chemisorption may be through the formation of Mn-S bonds. Moreover, more lattice oxygen was released during the reduction of Mn, which was involved in the further catalytic oxidation of S-containing species to produce SO_3_^2−^ and SO_4_^2−^, and may further form new oxygen vacancies to support more active sites. Although β-MnO_2_ enjoys a higher AOS based on the proportion of high valence Mn, α-MnO_2_ showed better catalytic activity due to more oxygen vacancies and stronger oxygen mobility.

## 4. Conclusions

In this paper, MnO_2_ catalysts with similar morphology but different crystal structures (α-MnO_2_ and β-MnO_2_) were successfully prepared, and the effects of the physicochemical properties on the catalytic activities were systematically investigated. Both Mn-based catalysts showed significant removal of CH_3_SH at 150 °C, achieving complete conversion, whereas α-MnO_2_ exhibited significantly better catalytic activity compared to β-MnO_2_ at a lower temperature (T < 100 °C) under a GHSV of 9000 mL g^−1^ h^−1^. Coupled with O_2_-TPD, H_2_-TPR, Raman spectra, XPS, EPR, and HRTEM, it was demonstrated that α-MnO_2_ has more oxygen vacancies, stronger surface oxygen migration ability, and better redox properties, which can be favorable for CH_3_SH decomposition. The readily released lattice oxygen during the reaction promoted further oxidative decomposition of S-containing species. The intermediate products of the solid and gas phases were determined as CH_3_SSCH_3_ and the S^0^, SO_3_^2−^, and SO_4_^2−^, respectively. The catalytic mechanism was further proposed as the oxygen vacancies on MnO_2_ provided active sites for the adsorption of CH_3_SH, facilitating the electron transfer of MnO_2_ with CH_3_SH, and the oxygen species derived from the Mn surface were further involved in the CH_3_SH catalytic oxidation. The findings of this study are essential for broadening the application of Mn-based catalysts in the removal of S-VOCs and providing new insights into the mechanism of interfacial reactions between VOCs and metal-based catalysts.

## Figures and Tables

**Figure 1 nanomaterials-13-00775-f001:**
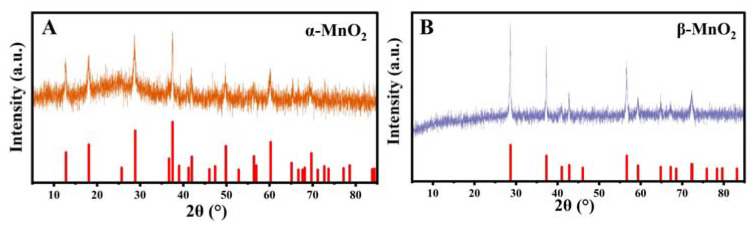
XRD patterns of α-MnO_2_ (**A**) and β-MnO_2_ (**B**).

**Figure 2 nanomaterials-13-00775-f002:**
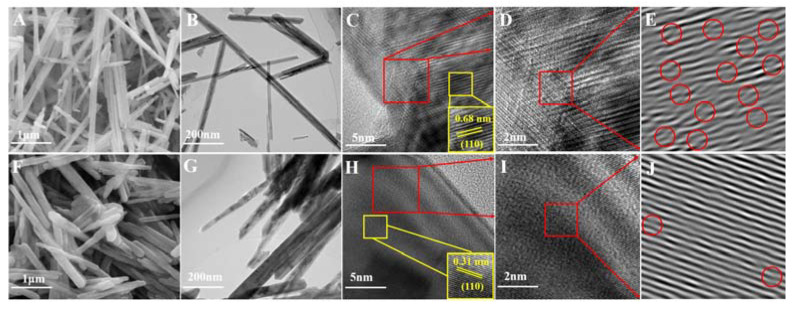
SEM, TEM and HRTEM images of α-MnO_2_ (**A**–**D**) and β-MnO_2_ (**F**–**I**); The inverse FFT images for α-MnO_2_ (**E**) and β-MnO_2_ (**J**).

**Figure 3 nanomaterials-13-00775-f003:**
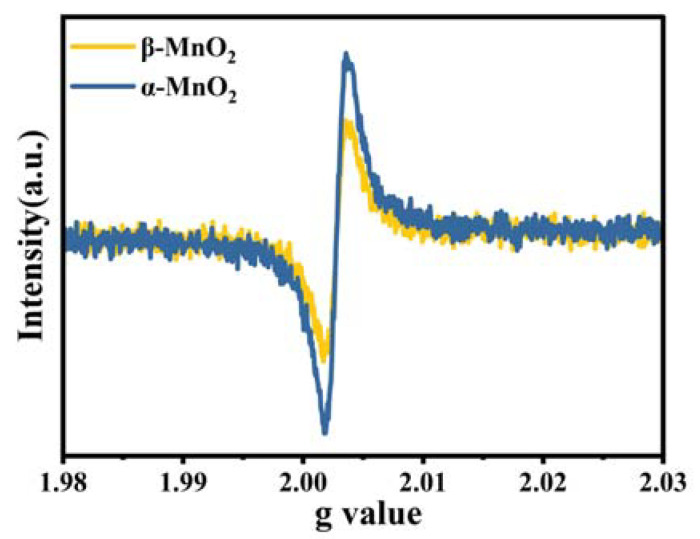
EPR profiles of α-MnO_2_ and β-MnO_2_.

**Figure 4 nanomaterials-13-00775-f004:**
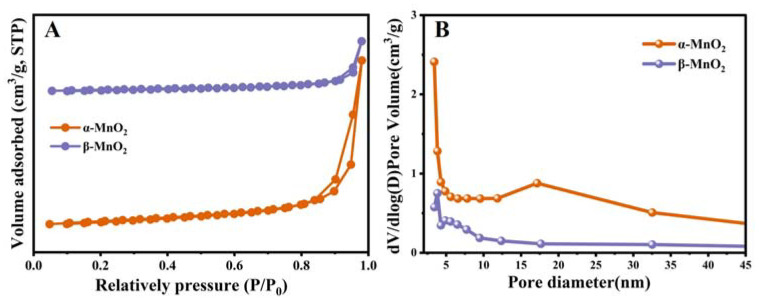
N_2_ adsorption-desorption isotherm plots (**A**) and pore distributions (**B**) of α-MnO_2_ and β-MnO_2_.

**Figure 5 nanomaterials-13-00775-f005:**
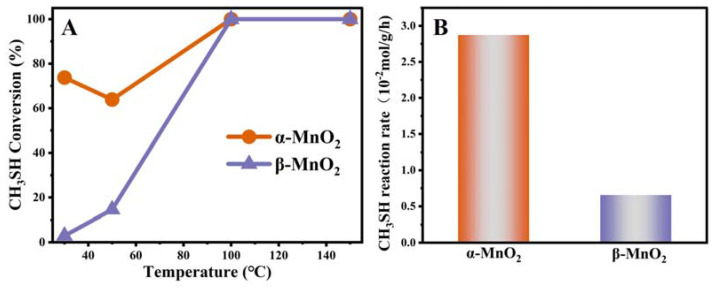
CH_3_SH degradation over α-MnO_2_ and β-MnO_2_ catalysts (**A**). Reaction condition: 0.20 g catalysts, 5000 ppm CH_3_SH, total flow rate = 30 mL min^−1^, WHSV = 9000 mL g^−1^ ·h^−1^); The reaction rates (×10^−2^ mol g^−1^ h^−1^) at 50 °C for α-MnO_2_ and β-MnO_2_ (**B**).

**Figure 6 nanomaterials-13-00775-f006:**
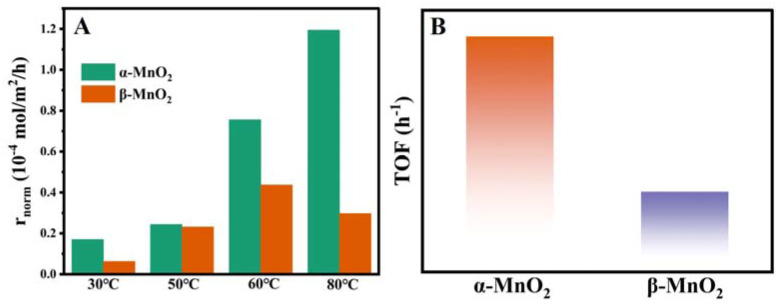
The value of r_norm_ (×10^−4^, mol·m^−2^ h^−1^) from 30 to 80 °C for α-MnO_2_ and β-MnO_2_ (A); The value of TOF (h^−1^) at 50 °C for α-MnO_2_ and β-MnO_2_ (B). Reaction conditions: 0.01 g catalysts, 5000 ppm styrene, total flow rate = 30 mL min^−1^, GHSV = 9000 h^−1^.

**Figure 7 nanomaterials-13-00775-f007:**
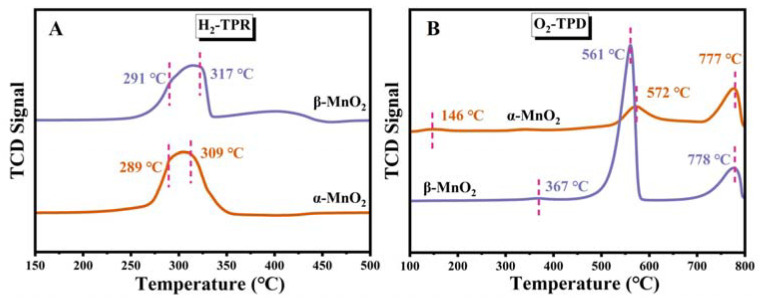
H_2_-TPR (**A**) and O_2_-TPD (**B**) profiles α-MnO_2_ and β-MnO_2_.

**Figure 8 nanomaterials-13-00775-f008:**
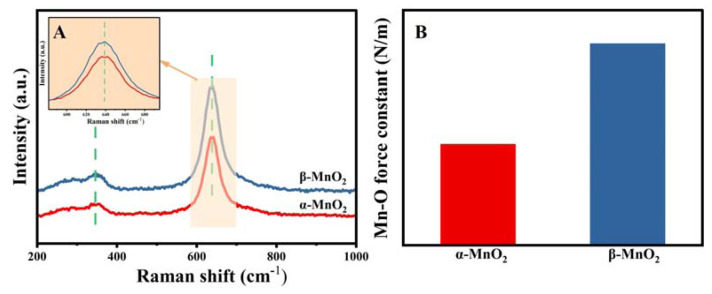
Raman profiles (**A**) and the Mn-O force constant (**B**) of α-MnO_2_ and β-MnO_2_.

**Figure 9 nanomaterials-13-00775-f009:**
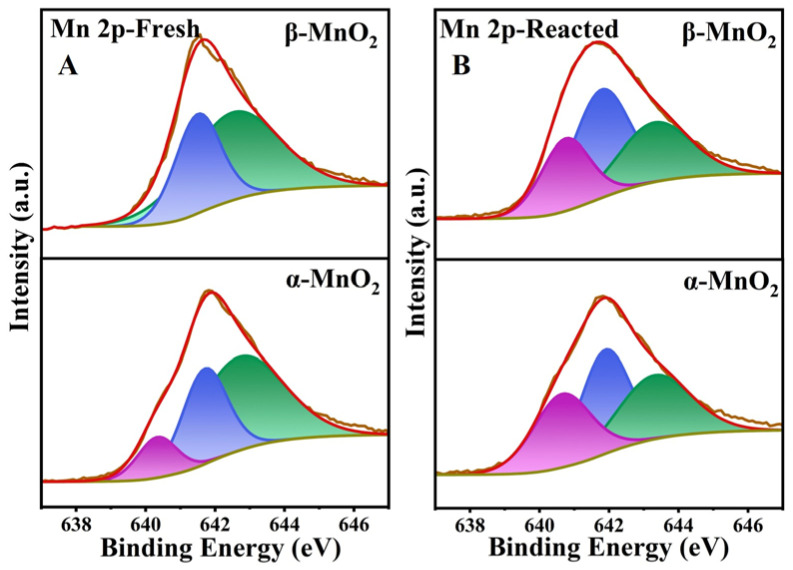
XPS profiles of Mn 2p_3/2_ over the fresh (**A**) and spent (**B**) catalysts.

**Figure 10 nanomaterials-13-00775-f010:**
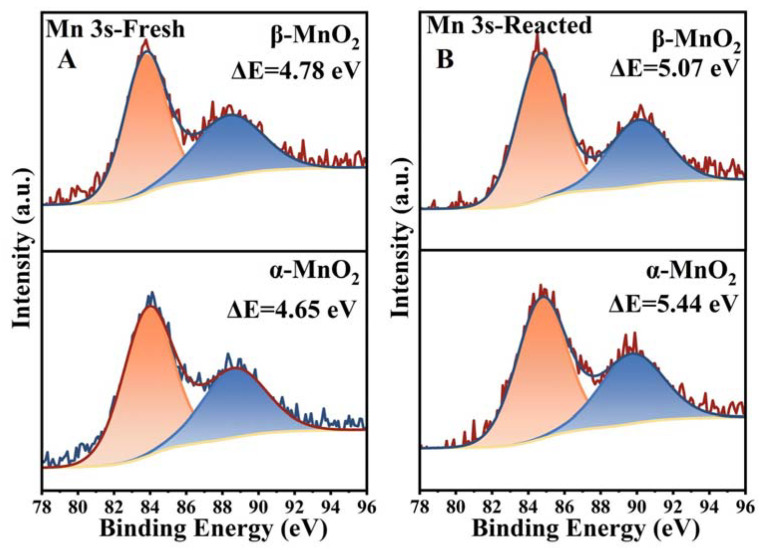
XPS profiles of Mn 3s over the fresh (**A**) and spent (**B**) catalysts.

**Figure 11 nanomaterials-13-00775-f011:**
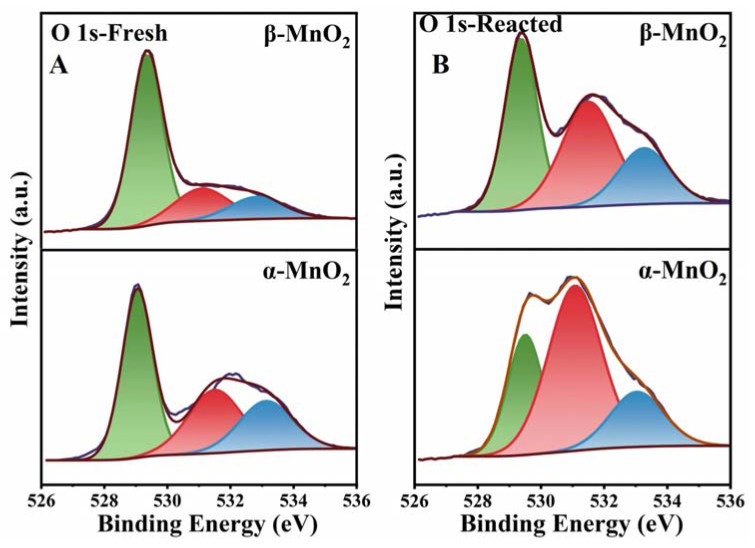
XPS profiles of O 1s over the fresh (**A**) and spent (**B**) catalysts.

**Figure 12 nanomaterials-13-00775-f012:**
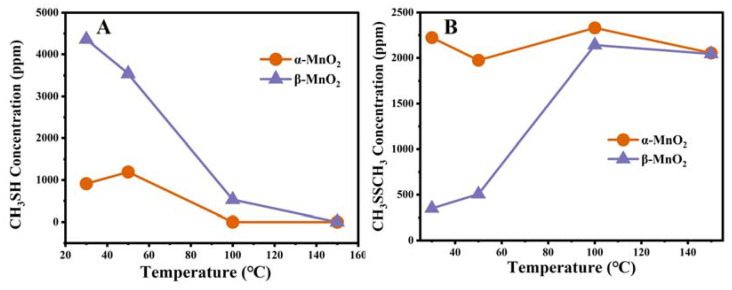
The CH_3_SH (**A**) and CH_3_SSCH_3_ (**B**) concentration on α-MnO_2_ and β-MnO_2_.

**Figure 13 nanomaterials-13-00775-f013:**
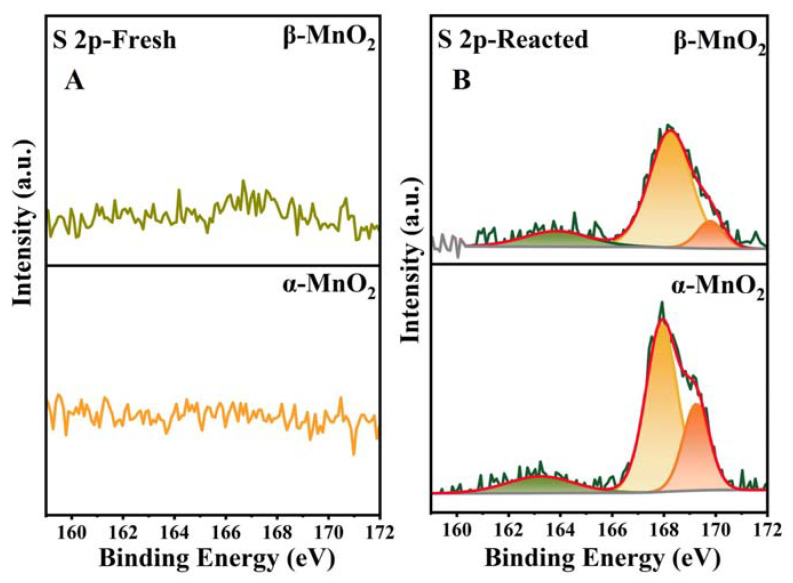
XPS profiles of S 2p over the fresh (**A**) and spent (**B**) catalysts.

**Figure 14 nanomaterials-13-00775-f014:**
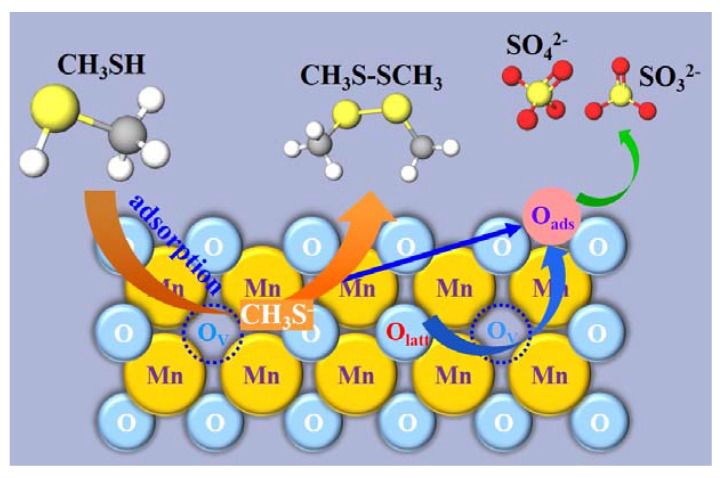
Reaction mechanism of CH_3_SH decomposition over MnO_2_ catalysts.

**Table 1 nanomaterials-13-00775-t001:** Mn 3s Mn 2p_3/2_ results before and after the reaction of B-MnO_2_ and C-MnO_2_.

Catalysts	Mn^2+^ (%)	Mn^3+^ (%)	Mn^4+^ (%)	Mn^2+^ + Mn^3+^/ Mn^4+^ (%)	AOS ^1^
α-MnO_2_-Fresh	11.31	30.31	58.38	0.71	3.57
β-MnO_2_-Fresh	2.71	35.04	62.26	0.6	3.75
α-MnO_2_-Spent	25.19	43.77	31.04	2.22	2.83
β-MnO_2_-Spent	16.39	46.99	36.62	1.99	3.25

^1^ AOS = 8.956 − 1.126 × ΔE.

**Table 2 nanomaterials-13-00775-t002:** O 1s results before and after the reaction of B-MnO_2_ and C-MnO_2_.

Catalysts	O_ads_	O_H2O_	O_latt_	O_ads_/O_latt_
α-MnO_2_-Fresh	30.22	23.53	46.25	0.65
β-MnO_2_-Fresh	20.68	14.10	65.22	0.32
α-MnO_2_-Spent	55.20	18.39	26.41	2.09
β-MnO_2_-Spent	39.78	20.34	39.87	1.00

**Table 3 nanomaterials-13-00775-t003:** S 2p results before and after the reaction of B-MnO_2_ and C-MnO_2_.

Catalysts	S^0^	S^4+^	S^6+^
α-MnO_2_-Fresh	0	0	0
β-MnO_2_-Fresh	0	0	0
α-MnO_2_-Spent	12.96	60.73	26.31
β-MnO_2_-Spent	13.38	74.06	12,56

## Data Availability

All date used to support the findings of this study are included within the article.

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
