# Peer review of "Comparative Study of α- and β-MnO2 on Methyl Mercaptan Decomposition: The Role of Oxygen Vacancies"

_nanomaterials, 2023, doi:10.3390/nano13040775_

Round 1

Reviewer 1 Report

The work from Hong Su et al describes the synthesis and catalytic activity of MnO2 nanorods with different crystalline structure, alpha and beta phases. The authors observed a higher activity for the alpha phase and related it to the higher oxygen vacancy concentration. The authors prove the presence of these defects with different techniques, in particular I believe and suggest them to revise this part. They show frequency filtered maps obtained from HRTEM images indicating the presence of these defects, I believe the images they show are not sufficient to prove the presence of the defects therefore they should improve the corresponding section.

Reviewer 2 Report

It is an interesting paper on the synthesis of α-MnO2 and β-MnO2 examined in the decomposition of methyl mercaptan. The catalysts were fully characterized by nitrogen adsorption/desorption, XRD, XPS, HRTEM, SEM, H2-TPR, O2-TPD and Raman spectroscopy. It is a meaningful work. Nevertheless, I suggest making some minor corrections before its publication in Nanomaterials:

1.  What amount of water was used for the washing of catalysts after centrifugation? Please add this detail in Materials and Methods Section.

2. The lines 167, 171: styrene should be replaced by methyl mercaptan.

3. The line 185: The XRD peaks assigned to β-MnO2 are wrong (they belong to α-MnO2 structure). Please add the correct ones.

Round 2

Reviewer 1 Report

I thank the authors for describing better the oxygen defects evaluation.